# Associations between neighborhood social capital, oral health risk factors, and tooth decay among Medicaid-enrolled adolescents: A hypothesis-generating preliminary study

Courtney M. Hill[1], Lloyd A. Mancl[1], Richard M. Carpiano[2], Adam C. Carle[3,4], Marilynn Rothen[1], Kyle Crowder[5], Michael Yoo[1], Donald L. Chi[1]*

1 Department of Oral Health Sciences, University of Washington, Seattle, Washington, United States of America, 2 School of Public Policy, University of California, Riverside, California, United States of America, 3 Department of Pediatrics, Cincinnati Children's Hospital Medical Center, University of Cincinnati College of Medicine, Cincinnati, Ohio, United States of America, 4 Department of Psychology, University of Cincinnati College of Arts and Sciences, Cincinnati, Ohio, United States of America, 5 Department of Sociology, University of Washington, Seattle, Washington, United States of America

* dchi@uw.edu

## Abstract

Neighborhood-based social capital – defined as resources within neighborhood social networks – is a potential contributor to adolescent oral health, but mechanisms that link the two are not well elucidated. We evaluated the potential mediating role of neighborhood, household, and individual oral health risk factors in the neighborhood social capital-tooth decay relationship. We collected cross-sectional data from 331 Medicaid-enrolled adolescents (ages 12–18 years) and one of their caregivers from 73 census tracts (neighborhoods) in three counties in Oregon, U.S.A in 2015 and 2016. Medicaid is a public insurance program in the U.S. providing no-cost dental insurance to low-income children. We measured four neighborhood social capital constructs: social support, social leverage, informal social control, and neighborhood organization participation. Oral health risk factors included worrying about food money, poor access to vegetables and fruits, inconsistent family and oral health routines, and adolescent stress. The outcome was number of untreated decayed tooth surfaces. Causal mediation analyses with mixed effect models were used to examine associations. Neighborhoods with higher social support had a lower prevalence of worrying about food money (prevalence ratio [PR] 0.74;95% CI: 0.56, 0.96;p = .02) as did neighborhoods with higher informal social control (PR 0.75;95% CI:0.58, 0.97;p = .03). All oral health risk factors were strongly associated with untreated decayed tooth surfaces. No form of neighborhood social capital was significantly associated with tooth decay. Natural indirect effects of neighborhood social support and informal social control operating through worrying about food money were not statistically significant. Future longitudinal studies that include robust measures

**Data availability statement:** All relevant data are within the paper and itsthe Supporting Information files.

**Funding:** This work was supported by the William T. Grant Foundation Scholars Award (DLC). https://wtgrantfoundation.org/funding/william-t-grant-scholars-program Services of the UW School of Dentistry's Regional Clinical Dental Research Center (RCDRC) were supported by the Institute of Translational Health Sciences, which is funded by the National Center For Advancing Translational Sciences of the National Institutes of Health under award number UL1TR00231 (DLC). https://ncats.nih.gov/ The funders had no role in study design, data collection and analysis, decision to publish, or preparation of the manuscript.

**Competing interests:** The authors have declared that no competing interests exist.

of neighborhood social capital and adequate sample sizes are needed to enable neighborhood-based interventions that promote adolescent oral health.

## Introduction

Tooth decay is one of the most common chronic diseases in the U.S. [1–4] and particularly for adolescents. At least one-half of the U.S. population aged 12–19 years has treated or untreated tooth decay, according to the most recent national estimates from 2015–2016 [5]. When untreated, its consequences include pain, sepsis, missed school days, decreased work efficiency, reduced quality of life, and other health conditions [6].

The causes of tooth decay in adolescents are multifactorial, resulting from biological, behavioral, environmental, and socioeconomic determinants [3] with socioeconomic disparities in the prevalence and burden of tooth decay being well-documented [4]. Yet, little existing research has examined the interplay of such factors, especially the social contexts that may serve as contributors to more proximal behavioral and biological pathways.

The present study seeks to formulate an empirical examination of how the community life of the neighborhoods in which adolescents reside contributes to these oral health risks. Social capital is defined as the resources accessible to members of a social group or network, such as neighborhood residents that exist within social networks [7]. Recognized as a determinant of health [8] and of oral health, specifically [9–11], an improved understanding of how social capital influences the development of common diseases like tooth decay could help inform public health efforts aimed at addressing oral health disparities. Previous research has revealed four forms of social capital: (1) social support (provisions that help residents cope with everyday challenges), (2) social leverage (sharing information on health- and non-health-related issues), (3) informal social control (maintenance of safety and norms), and (4) neighborhood organization participation (organized efforts that address community quality of life and personal well-being) [7,12].

International studies have reported mixed findings regarding the impact of social capital on clinical and subjective oral health outcomes among children, adolescents, and adults [10,11]. These findings are driven, in part, by inconsistencies in how social capital is operationalized, including an individual-level and a contextual-level conceptualization (e.g., at the community, neighborhood, or household level) [10]. For adolescents in East London, higher individual perceived social support was associated with fewer decayed tooth surfaces [13]. Similarly, among adolescents in Brazil, individual participation in community groups [14] and higher individual perceived social support [15] were each associated with fewer decayed tooth surfaces and lack of individual or community social capital was associated with lower oral health related quality of life [16]. In contrast, a study of Japanese students ages 18–19 years reported high neighborhood informal social control was associated with poorer self-rated oral health [17]. Despite these inconsistencies, a systematic review of 31

studies examining social capital and oral health among children and adolescents found community-level social capital had more influence than individual-level social capital, and associations were stronger for subjective oral health measures versus clinical oral health measures [11].

Although theories exist for how community social capital affects health, few empirical studies to date have identified potential mechanisms that link social capital to health outcomes like tooth decay. The lack of mechanism knowledge is a barrier in developing social capital-based interventions. Certain known oral health risk factors—including food insecurity [18–20], stress [21–23], poor dietary patterns with little vegetable and fruit intake [24–27], inconsistent family [28] and oral health routines [29]—are candidate mediators, as they are plausibly affected by neighborhood social capital. For instance, studies have documented an inverse association between neighborhood social capital and household food insecurity [30–32], which can be explained by food-insecure community members' increased ability to draw on support from neighbors to access instrumental resources like food and groceries, car rides to the grocery store, or money for shopping. In this way, some neighborhood-level conditions may mediate the effects of household-level behaviors. Neighborhood social support can also manifest as increased emotional support to help cope with daily problems, thereby minimizing the negative effects of stress. Similarly, in a neighborhood with high levels of organization participation, community advocacy efforts can secure better resources in the neighborhood like access to healthier food stores, fluoridated tap water, and community clinics that provide comprehensive dental care services. In neighborhoods with high informal social control, adherence to social norms that include healthy behaviors (i.e., healthy eating versus sugary snacks and fast food family; routines like eating meals together every day; or enforcement of oral hygiene behaviors like regular tooth brushing and dental care) may also improve oral health outcomes. There are also theorized mechanisms by which neighborhood social capital could lead to negative health outcomes [33,34]. For example, a systematic review documented at least two negative consequences of social capital including behavioral contagion and cross-level interactions between social cohesion and individual characterisitcs [32].

In this hypothesis-generating study, we begin to address the aforementioned knowledge gaps by evaluating (a) associations between neighborhood social capital and oral health risk factors among publicly-insured adolescents in the U.S. and (b) the extent to which oral health risk factors mediate the social capital-tooth decay relationship. The study contributes to efforts examining social capital as a determinant of oral health and aims to identify potential mediators of the relationship.

## Methods

### Preliminary conceptual model

Our analyses are guided by a preliminary conceptual model (Fig 1) that hypothesizes that household and behavioral oral health risk factors mediate the relationship between neighborhood social capital and adolescent tooth decay. Neighborhood income and rurality were identified as neighborhood-level confounders of the associations. Household-level confounders were identified as household income, employment, and home ownership/renting. Individual-level confounders were identified as child age, child sex, child race, and child ethnicity.

### Study design and population

Our multilevel study [35]. was based on cross-sectional data collected from low-income adolescents, ages 12–18 years, who resided in one of 73 neighborhoods (defined as census tracts) located within three counties (Multnomah, Tillamook, or Hood River) of the western U.S. state of Oregon. These counties were selected to include adolescents from both urban and rural areas of the state, with specific neighborhoods restricted to those with at least 50 Medicaid-enrolled adolescents based on Medicaid enrollment files, to increase the likelihood of recruiting sufficient numbers of participants for clinical data collection. Medicaid is a state and federal program that provides no-cost health insurance for low-income enrollees

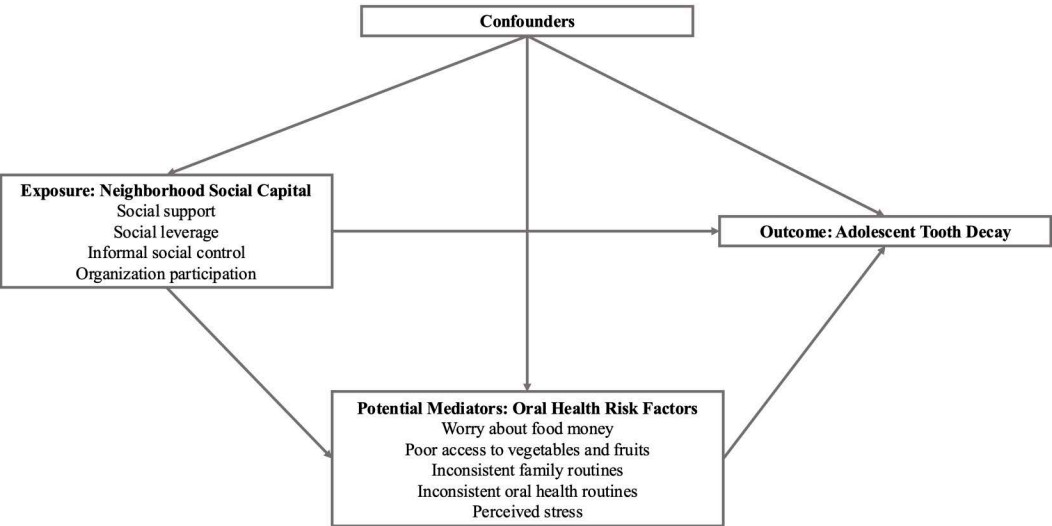

**Fig 1. Conceptual model for neighborhood social capital, oral health risk factors, and tooth decay among Oregon Medicaid-enrolled adolescents, aged 12 to 18 years, who participated in a neighborhood oral health study in 2015-2016.** Footnotes: The conceptual model for the study hypothesized that household and behavioral oral health risk factors are mediators for the association between neighborhood social capital and tooth decay. Neighborhood income and rurality were identified as neighborhood-level confounders of the associations. Household-level confounders were identified as household income, employment, and home ownership/renting. Individual-level confounders were identified as child age, child sex, child race, and child ethnicity.

and is mandated to cover extensive dental services for all enrollees age 20 and under. Dental care includes comprehensive treatment to address pain and infection, restoration of teeth, and preventive services to maintain oral health [36]. Each adolescent's neighborhood was coded based on their residential address in the Medicaid enrollment files.

## Study recruitment

Eligible adolescents residing in the target study neighborhoods were recruited as previously described [37]. Briefly, study staff contacted caregivers of the eligible adolescents via telephone numbers listed in Oregon Medicaid enrollment files obtained from the Oregon Health Authority, confirmed eligibility through the adolescent's primary caregiver, and invited caregiver-adolescent dyads to participate in the study [37]. Interested dyads were scheduled to visit a local dental office in Oregon where consent, assent, enrollment, and data collection occurred at a single study visit. All study visits took place between December 2015 and December 2016 and lasted about 1 hour. The study was approved by the University of Washington Institutional Review Board.

## Study data

**Neighborhood social capital data collection.** To measure neighborhood social capital, we used a survey that was completed (a) by caregivers of adolescents in the study at the time of study enrollment and (b) a sample of neighborhood informants identified in the Oregon Medicaid enrollment files who were contacted by mail. Inclusion criteria for neighborhood informants included a valid mailing address within a study neighborhood and evidence of at least one household member enrolled in Medicaid. Paper surveys were mailed in both English and Spanish to a random saple of households along with a $2 bill as a thank you. On average, 94 surveys were sent to informants in each of the 73 study neighborhoods (range 17–315) with 7,397 surveys mailed in total, of which 721 were returned. The mean neighborhood

response rate was 11% (range 2% to 42%). Demographic characteristics of participating neighborhood informants were not collected.

**Adolescent data collection.** Caregivers and adolescents each completed a paper survey and adolescents received a dental screening. The surveys included questions on oral health risk factors and adolescent demographic characteristics. A trained and calibrated pediatric dentist or dental hygienist completed the dental screenings. All erupted and visible primary and permanent tooth surfaces were classified as decayed, missing, filled, or non-cavitated white spot lesions based on the National Institute for Dental and Craniofacial Research (NIDCR) Early Childhood Caries Collaborating Centers (EC4) Criteria [38].

## Study variables

**Neighborhood social capital.** Neighborhood social capital was assessed via 25-items (S1 Table in S1 File). Four forms of neighborhood social capital were created via empirical Bayes residuals derived from multilevel linear regression models—an approach common in social capital research [12]. The final four forms were social support, social leverage, informal social control, and neighborhood organization participation. The value for each social capital variable represents the deviation from the neighborhood-level grand mean score for that variable with higher scores representing higher social capital.

**Oral health risk factors.** We examined five oral health risk factors: worrying about food money running out, poor neighborhood access to vegetables and fruits, inconsistent family routines, inconsistent oral health routines, and adolescent stress.

*Worrying about food money running out* was used as a measure of food insecurity and assessed with a single item using the "last 12 months" as the reference period: "We worried whether our food would run out before we got money to buy more". Response options were never true, sometimes true, or often true. The item was dichotomized (never or some-times vs. often) when modeled as an outcome.

*Poor neighborhood access to vegetables and fruits* was measured with a single item: "A large selection of fruits and vegetables is available in my neighborhood". Five response options ranged from strongly disagree to strongly agree. We considered responses of "disagree" or "strongly disagree" to indicate the respondent has poor neighborhood access to vegetables and fruits.

Two types of routines were assessed: inconsistent general *family* routines and inconsistent *oral health* routines. We measured the former using a modified version of the validated Family Routine Inventory (FRI) which measures cohesion, solidarity, order, and overall satisfaction with family life [39]. The FRI contains 26 items. Three items were not applicable to adolescents (e.g., "parents read or tell stories to the children almost every day") and thus removed. Response options were almost never, 1–2 times a week, 3–5 times a week, always/everyday. A composite score was created by summing the responses for the remaining 23 items and was reverse-coded so larger values indicated lower endorsement and adherence to a routine. The range of possible scores was 0–69 and missing item-level data were counted as 0.

To measure inconsistent *oral health routines*, we created three additional items as an extension of the FRI: "How important is making sure children brush their teeth after he/she wakes up in the morning?"; "How important is making sure children brush his/her teeth before bedtime?"; and "How important is making sure children see a dentist regularly for checkups?". These items had the same response options as the aforementioned FRI items. The range of possible composite scores was 0–9 and the composite score was reverse-coded so a larger value indicated lower endorsement and adherence to a routine, and missing item-level data were counted as 0.

*Adolescent stress* was measured using the self-reported 10-item perceived stress scale (PSS), which has been pre-viously validated in adolescent populations [40,41]. For each item, respondents were asked how often they felt a certain way on a five-point scale from never to very often with never scored as 0 and very often scored as 4. A total PSS score

is the sum across all items, with higher scores indicating higher levels of perceived stress with a range of possible values from 0 to 40, missing item-level data were counted as 0.

**Tooth decay.** Untreated tooth decay was the primary study outcome. It reflects current disease status (i.e., including missing teeth and filled surfaces may capture disease that occurred before the exposures of interest in this study and were therefore not part of the primary outcome). It is measured as the number of untreated decayed tooth surfaces on primary or permanent teeth. We also evaluated the number of decayed, missing, and filled tooth surfaces as a secondary outcome to allow for comparison of our findings with previous social capital and adolescent tooth decay research [13–15].

**Confounders.** We adjusted for multiple neighborhood and individual confounders that were available for the study population, including neighborhood median income, neighborhood rurality, child age, and child sex. Neighborhood median income was obtained from the 2012–2016 American Community Survey [42] and was treated as a continuous variable when adjusted for as a confounder. Neighborhood rurality was based on the 2010 Rural-Urban Commuting Area (RUCA) codes which categorize census tracts based on population density, urbanization, and commuting [43]. We used categorization D which defines "urban" as all places that have 30% or more of their workers going to a Census Bureau-defined "Urbanized Area" [44]. According to the RUCA classification, all neighborhoods in Hood River and Tillamook counties were considered rural and all neighborhoods in Multnomah county were considered urban. Models were not adjusted for other household- or individual-level confounders including household income, employment, and child race and ethnicity because these variables were either not measured or had high levels of missingness. To describe the study population, we also measured whether adolescents had irregular dental care use, by asking caregivers on the caregiver survey, "Does your child currently see a dentist for regular checkups?" A "no" response was considered irregular dental care.

## Statistical analyses

We generated descriptive statistics for participant demographic characteristics and used generalized linear mixed-effects model (GLMM) to model study associations. For associations between neighborhood social capital and oral health risk factors we used log binomial regression for binary oral health risk factors and linear regression for continuous oral health risk factors. We then used Poisson regression to examine associations between oral health risk factors and tooth decay. Lastly, we used mediation analysis based on the potential outcomes framework [45] to estimate natural direct and indirect effects for the relationship between neighborhood social capital and tooth decay. For all analyses, a random effect for study neighborhood was used to account for non-independence at the neighborhood level and each estimate was reported with a 95% confidence interval (95% CI).

For the mediation analysis, we fit a Poisson regression model with robust standard errors [46] for the mediator conditional on neighborhood social capital and confounders [45,47–49]. We then fit a Poisson regression model for number of untreated decayed tooth surfaces on neighborhood social capital, the mediator, and confounders.

We estimated two mediation effects: 1) the natural indirect effect (NIE), which measures the effect of the exposure that operates through the mediator; and 2) the natural direct effect (NDE), which measures the effect of the exposure if it did not cause the mediator [50]. The proportion mediated was estimated and expressed as a percentage. The proportion mediated can be interpreted as the overall effect of social capital on untreated tooth decay attributable to the impact of social capital on the mediator.

All estimates for the mediation analysis were generated using the quasi-Bayesian approach developed by Imai et al. that uses Monte Carlo simulations based on normal approximations [45,47–49]. In this analysis, we used 1,000 simulations. Estimated effects for the mediation analyses were reported on the additive scale (mean differences in the number of untreated decayed tooth surfaces) and estimated at the mean level of confounders, or the most frequent level of categorical confounders. Consistent with mediation analysis reporting guidelines [51], we examined whether including an interaction term between the exposure and mediator changed the natural effects in the mediation analysis.

All models were adjusted for available confounders: neighborhood median income, neighborhood rurality, child age, and sex. An alpha value of 0.05 was considered statistically significant. All analyses were conducted in R version 4.03. Mediation analyses were completed using the mediation package in R [49] and the lme4 package was used for mixed-effects models [52].

## Results

### Descriptive statistics

The mean age of 331 study participants was 15 years (standard deviation [SD]=1), 53% were female, and 24% had irregular dental care (Table 1). While 72% of the adolescents for whom we had race/ethnicity data available were white, race/ethnicity was missing for 50% of the sample. The median annual neighborhood income was about $55,000 and 82% of adolescents lived in an urban neighborhood. In terms of oral health risk factors, 41% of the sample reported sometimes worrying about food money running out and 7% reported often worrying about food money. Nine-percent reported poor neighborhood access to vegetables and fruits. The mean family routines score was 22 (SD = 11), the mean oral health routines score was 2 (SD = 2), and the mean perceived stress score was 17 (SD = 7). The mean number of untreated decayed tooth surfaces was 0.99 (SD = 2.84; median = 0.00; interquartile range = 0.00,1.00).

### Neighborhood social capital and oral health risk factors

Thirty-two adolescents were excluded because of any missing data. The final analytic sample consisted of 299 adolescents. After adjusting for confounders, neighborhoods with higher social support had a lower prevalence of worrying about food money (prevalence ratio [PR] 0.74; 95% CI 0.56, 0.96; p = .02) as did neighborhoods with higher informal social control (PR 0.75; 95% CI 0.58, 0.97; p = .03) (Table 2). This means that households in neighborhoods with higher social support reported on average were 26% less likely to experience worry about food money (a form of food insecurity). No forms of neighborhood social capital were associated with other oral health risk factors including poor neighborhood access to vegetables and fruits, inconsistent family routines, inconsistent oral health routines, or perceived stress (p > .05).

### Oral health risk factors and tooth decay

All oral health risk factors were positively and significantly associated with the mean number of untreated decayed tooth surfaces (Table 3). After adjusting for confounders, adolescents in households that often worried about food money had 2.61 times the untreated decayed tooth surfaces compared to adolescents in households that never worried about food money (95% CI 1.60, 4.24; p < .001). Similarly, poor neighborhood access to vegetables and fruits was associated with 1.46 times the untreated decayed tooth surfaces (95% CI 1.03, 2.06; p = .03). Lower endorsement and adherence to family routines and oral health routines were each associated with higher levels of untreated decayed tooth surfaces (mean ratio [MR] 1.03; 95% CI 1.02, 1.04; p < .001 and MR 1.28; 95% CI 1.22,1.35; p < .001, respectively). A one-unit higher PSS score was associated with 1.07 times the untreated decayed tooth surfaces (95% CI 1.04, 1.09; p < .001).

### Mediating effects of worrying about food money

Because worrying about food money was the only oral health risk factor significantly associated with multiple forms of neighborhood social capital, we chose to evaluate it as a potential mediator in the hypothesized neighborhood social capital–tooth decay relationship. Including an interaction between the exposure and mediator in the main mediation analysis did not change the NIE and NDE estimates (S2 Table in S1 File). Total effect estimates indicated that higher neighborhood social support and informal social control were not associated with higher levels of untreated decayed tooth surfaces (mean difference (MD) 0.04; 95% CI −0.37, 0.48; p = .86 and MD 0.24; 95% CI −0.09, 0.64; p = .16, respectively) (Table 4). The NIEs of neighborhood social support and informal social control operating through worrying about food money were

**Table 1. Characteristics of Oregon Medicaid-enrolled adolescents ages 12 to 18 years who participated in a neighborhood oral health study in 2015-2016 (N = 331).**

|  | Mean±SD, range or N (%) |
|---|---|
| **Decayed tooth surfaces** | 0.99 ± 2.84 |
| Median (IQR) | 0.00 (0.00, 1.00) |
| Missing | 3 |
| **Adolescent age (years)** | 15 ± 1, 12-18 |
| Missing | 25 |
| **Adolescent sex** |  |
| Female | 161 (53%) |
| Male | 145 (47%) |
| Missing | 25 |
| **Adolescent race/ethnicity[1]** |  |
| American Indian or Alaska Native | 2 (1%) |
| Asian or Pacific Islander | 12 (7%) |
| Black | 20 (12%) |
| Hispanic | 12 (7%) |
| White | 118 (72%) |
| Missing | 167 |
| **Irregular dental care[2]** | 78 (24%) |
| Missing | 6 |
| Neighborhood median income[3] (IQR), US$ | $54,891 ($29,677-$86,892) |
| **Neighborhood rurality** |  |
| Urban | 273 (82%) |
| Rural | 58 (18%) |
| **Worrying about food money running out** |  |
| Never true | 169 (52%) |
| Sometimes true | 133 (41%) |
| Often true | 23 (7%) |
| Missing | 6 |
| **Poor neighborhood access to vegetables and fruits** | 31 (9%) |
| Missing | 6 |
| **Inconsistent family routines[4]** | 22 (11), 0-54 |
| Missing | 4 |
| **Inconsistent oral health routines[5]** | 2 (2), 0-9 |
| Missing | 4 |
| **Perceived stress[6]** | 17 (7), 0-40 |
| Missing | 1 |

IQR, interquartile range.

[1]Race/ethnicity, age, and sex data were obtained from Oregon Medicaid enrollment files which have a large amount of missingness for demographic characteristics.

[2]To measure whether adolescents had irregular dental care use, we asked caregivers on the caregiver survey, "Does your child currently see a dentist for regular checkups?" A "no" response was considered irregular dental care.

[3]Neighborhood median income was obtained from the 2012–2016 American Community Survey collected by the U.S. Census Bureau.

[4]The range of possible scores for family routines was 0–69.

[5]The range of possible scores for oral health routines was 0–9.

[6]Perceived stress is based on the 10-item Perceived Stress Scale and the range of possible scores was 0–40.

not statistically significant. Examining DMFS as the outcome for the main mediation analysis showed similar results to untreated tooth decay (S3 Table in S1 File).

## Discussion

The goal of this hypothesis-generating study was to explore associations between neighborhood social capital, oral health risk factors, and tooth decay prevalence for Medicaid-enrolled adolescents. We also conducted preliminary causal mediation analyses to identify hypotheses for future mediation studies on the neighborhood social capital–tooth decay relationship. Two forms of social capital, neighborhood social support and informal social control, were inversely associated with the prevalence of worrying about food money. No form of neighborhood social capital was statistically significantly associated with untreated tooth decay and mediation of neighborhood social support and informal social control operating through worrying about food money were not statistically significant. In summary, we found that worrying about food money, a component of food insecurity and known risk factor for poor oral health, may be a relevant mediator for the neighborhood social capital–tooth decay relationship but further investigations are needed.

### Neighborhood social capital and oral health risk factors

Among Medicaid-enrolled adolescents in the study, neighborhood social support and informal social control were associated with lower prevalence of worrying about food money, a contributor to food insecurity. The connection between social support and food insecurity has been previously reported in the literature [30–32,53]. A recent review reported that higher levels of social capital improve multiple aspects of food security including food availability, accessibility, utilization, and food system stability – through the mechanism of food sharing and information sharing between community members [54]. Less is known about the association between neighborhood informal social control and food insecurity. Qualitative research suggests that behaviors and priorities around food purchasing and diet are influenced by social control [55] and at least one longitudinal study has reported that social control influences food insecurity through neighborhood cohesiveness [56]. Food insecurity is a well-documented risk factor for poor oral health, especially tooth decay [18–20], and findings from this study provide further support for the food insecurity-tooth decay relationship for Medicaid-enrolled adolescents. Given the evidence of an association between neighborhood social capital and food insecurity and the well-documented association between food insecurity and tooth decay, food insecurity is a promising potential mediator in a neighborhood social capital—oral health relationship. However, further research is needed on populations beyond Medicaid-enrolled adolescents to evaluate the generalizability of these findings.

We did not detect statistically significant associations between neighborhood social capital and other oral health risk factors examined in this study, including poor neighborhood access to vegetables and fruits, inconsistent family and oral health routines, and perceived stress. The lack of significant findings may reflect how we measured these variables and the null findings should be interpreted cautiously due to our underpowered sample size. Informal social control can be understood as the adherence to social norms that promote healthy behaviors. In the context of oral health, healthy behaviors that manifest through family routines could include consistent and shared meal times. Eating breakfast regularly has been associated with better oral health outcomes, with research in the U.S. suggesting that not eating breakfast daily is associated with almost four times the odds of tooth decay in primary teeth for children ages 2–5 years [24]. Neighborhood informal social control could also plausibly impact oral health routines, although, we did not detect an association between neighborhood informal social control and oral health routines (measured as regular toothbrushing and dental visits) in this study. Theory has also pointed to stress as a pathway for neighborhood social capital to influence health and while we found that adolescents' perceived stress was positively associated with the number of decayed tooth surfaces in this study, no form of neighborhood social capital was statistically significantly associated with stress. More work is needed to refine how these forms of social capital are specifically related to oral health. For example, one possibility is to examine

**Table 2. Associations of neighborhood social capital and oral health risk factors for Oregon Medicaid-enrolled adolescents ages 12 to 18 years who participated in a neighborhood oral health study in 2015-2016 (n = 299).**

| Neighborhood social capital[1] | Worrying about food money[2] | | | Poor neighborhood access to vegetables and fruits | | | Inconsistent family routines[3] | | | Inconsistent oral health routines[3] | | | Perceived stress[4] | | |
|---|---|---|---|---|---|---|---|---|---|---|---|---|---|---|---|
| | PR | 95% CI | p | PR | 95% CI | p | Beta | 95% CI | p | Beta | 95% CI | p | Beta | 95% CI | p |
| Social support | **0.74** | **0.56, 0.96** | **.02** | 0.83 | 0.64, 1.08 | .16 | −0.63 | −1.6, 0.38 | .22 | 0.06 | −0.15, 0.26 | .57 | −0.47 | −1.0, 0.09 | .10 |
| Social leverage | 1.07 | 0.84, 1.35 | .60 | 0.98 | 0.79, 1.20 | .83 | −0.36 | −1.1, 0.35 | .32 | −0.01 | −0.16, 0.13 | .87 | −0.15 | −0.57, 0.26 | .46 |
| Informal social control | **0.75** | **0.58, 0.97** | **.03** | 0.87 | 0.70, 1.08 | .21 | −0.73 | −1.5, 0.07 | .07 | 0.06 | −0.10, 0.23 | .45 | 0.01 | −0.45, 0.47 | .96 |
| Organization participation | 0.91 | 0.61, 1.35 | .63 | 1.24 | 0.91, 1.70 | .17 | −0.23 | −1.5, 1.0 | .72 | −0.18 | −0.43, 0.08 | .17 | −0.14 | −0.85, 0.57 | .70 |

PR, prevalence ratio; CI, confidence interval

[1]Neighborhood social capital measures are coded so higher values indicate higher social capital.

[2]Worry about food money was dichotomized (never or sometimes vs. often) when modeled as an outcome.

[3]Family routines and oral health routines are continuous variables where a larger value indicates lower endorsement and adherence to a routine.

[4]Perceived stress is based on the 10-item perceived stress scale and a higher score indicates a higher level of stress.

Log-binomial regression was used to estimate PR, 95% CI, and p-values for binary outcomes (worrying about food money, poor neighborhood access to vegetables and fruits) and linear regression was used to estimate betas, 95% CIs, and p-values for linear outcomes (family routines, oral health routines, perceived stress). All models were adjusted for neighborhood-level and individual-level confounders: neighborhood median income, neighborhood rurality, child age, and child sex. General linear mixed effect models with neighborhood was treated as a random effect to account for clustering.

**Table 3. Associations of oral health risk factors and tooth decay for Oregon Medicaid-enrolled adolescents ages 12 to 18 years who participated in a neighborhood oral health study in 2015-2016 (n = 299).**

| Oral health risk factor | Decayed tooth surfaces (mean) | Mean ratio | 95% CI | p |
|---|---|---|---|---|
| Worrying about food money | | | | |
| Never true | 0.99 | Ref | Ref | |
| Sometimes true | 1.08 | 1.13 | 0.86, 1.48 | .36 |
| Often true | 1.45 | 2.61 | 1.60, 4.24 | <.001 |
| Poor neighborhood access to vegetables and fruits | | | | |
| No | 1.03 | Ref | Ref | Ref |
| Yes | 1.45 | 1.46 | 1.03, 2.06 | .03 |
| Inconsistent family routines[1] | – | 1.03 | 1.02, 1.04 | <.001 |
| Inconsistent oral health routines[1] | – | 1.28 | 1.22, 1.35 | <.001 |
| Perceived stress[2] | – | 1.07 | 1.05, 1.09 | <.001 |

CI, confidence interval.

[1]Family routines and oral health routines are continuous variables where a larger value indicates lower endorsement and adherence to a routine.

[2]Perceived stress is based on the 10-item perceived stress scale and a higher score indicates a higher level of stress.

Poisson regression was used to estimate mean ratios, 95% CI, and p-values. All models were adjusted for neighborhood-level and individual-level confounders: neighborhood median income, neighborhood rurality, child age, and child sex. General linear mixed effect models with neighborhood was treated as a random effect to account for clustering.

social leverage in the context of whether neighborhood information about employment can be used to secure private dental care benefits [12].

### Impact of neighborhood social capital on adolescent tooth decay

Findings from mediation analysis suggested that neighborhood social support and neighborhood informal social control were not associated with tooth decay. This finding is inconsistent with previous studies that have reported protective

**Table 4. Mediating effects of worrying about food money between neighborhood social capital and tooth decay for Oregon Medicaid-enrolled adolescents ages 12 to 18 years who participated in a neighborhood oral health study in 2015-2016 (n = 299).**

| Neighborhood social capital[1] | Proposed mediator | Decayed tooth surfaces | | | | | | |
|---|---|---|---|---|---|---|---|---|
| | | Total effect (95% CI) | p | NIE[3] (95% CI) | p | NDE[4] (95% CI) | p | % Mediated |
| Social support | Worrying about food money[2] | 0.04 (−0.37, 0.48) | .86 | −0.03 (−0.21, 0.11) | .67 | 0.06 (−0.28, 0.50) | .76 | 8.3% |
| Informal social control | Worrying about food money[2] | 0.24 (−0.09, 0.64) | .16 | −0.03 (−0.22, 0.12) | .67 | 0.26 (0.03, 0.64) | .09 | 4.9% |

NIE, natural indirect effect; NDE, natural direct effect.

[1]Neighborhood social capital measures are coded so higher values indicate higher social capital.

[2]Worry about food money was dichotomized (never or sometimes vs. often) when modeled as an outcome.

[3]The NIE can be interpreted as the effect of neighborhood social capital on tooth decay that operates through the mediator.

[4]The NDE can be interpreted as the effect of neighborhood social capital on tooth decay if food insecurity did not cause the mediator.

Poisson regression was used to model worrying about food money and tooth decay. All models were adjusted for neighborhood-level and individual-level confounders: neighborhood median income, neighborhood rurality, child age, and child sex. Neighborhood was treated as a random effect to account for clustering. Estimated effects for the mediation analyses are reported on the additive scale (mean differences in the number of untreated decayed tooth surfaces) and estimated at the mean level of confounders, or the most frequent level of categorical confounders.

impacts of social capital on adolescent tooth decay [13–15]. However, the literature is mixed as negative impacts of social capital on oral health have also been reported. For instance, among first-year college students in Japan, high neighborhood informal social control was associated with worse self-rated oral health [17] and in a sample of adults in Los Angeles, California, U.S., residing in a neighborhood with higher social support was associated with lower odds of dental care use [12]. Underlying theory suggests that a "dark side" of social capital arises from exchange of antagonistic information and upholding of damaging social norms through behavioral contagion and cross-level interactions between social cohesion and individual characteristics [33,34]. In the case of the Japanese college student study, high informal social control was hypothesized to cause individuals to feel overly constrained resulting in stress [17].

One reason that we failed to find an association between neighborhood social support and tooth decay in this study may be attributable to differences in the social capital measures between previous work and this study. For studies reporting an inverse association between social capital and tooth decay among adolescents in East London and Brazil, individual-level measures of perceived social support and community participation were used, while our study focused on contextual measures at the neighborhood level [13–15]. This explanation is somewhat supported by the Japanese study that found harmful impacts of neighborhood social support and neighborhood informal social control since they also used neighborhood-level social capital measures [12,17]. However, another important consideration for the interpretation of this finding is our underpowered study sample size and potential bias because of residual confounding. These considerations should be taken into account when developing a longitudinal study to examine the association of neighborhood social capital and tooth decay.

## Mediating effects of worrying about food money

The natural indirect effects of neighborhood social support and informal social control operating through worrying about food money were not statistically significant. Given the relationships between neighborhood social capital and food insecurity and between food insecurity and tooth decay, the lack of significant mediation effects is somewhat surprising. However, null study findings may have resulted from an underpowered mediation analysis given that simulation studies

have reported that a sample size of at least 500 is needed to detect small mediation effects [57], like those that could be anticipated in this study. The only other known study to examine mediators in the neighborhood social capital–tooth decay relationship followed a cohort of Brazilian children for a decade and found that community-level social capital was negatively associated with incidence of tooth decay and the effect was mediated through sense of coherence, frequency of toothbrushing, and use of dental services [58]. Unlike the Brazilian study, our study failed to observe an association between neighborhood social capital and perceived stress and oral health routines. This could be because of the differences between sense of coherence and perceived stress. Sense of coherence captures how stress is managed and when under high stress, those with a high social capital may experience salutary effects [59]. Differences may also have resulted from different definitions of social capital. Specifically, community social capital in the Brazilian study was defined using objective measures including the presence of community cultural centers, number of dental workers, and number of churches in the neighborhood while our study used four forms of social capital which were constructed from surveys administered to neighborhood informants. Regardless, future work examining the proposed mediators in this study is warranted to enable the development of evidence-based interventions aimed at improving oral health.

## Strengths and limitations

Limitations should be considered when interpreting the findings of this study. First, it is possible that there are unmeasured confounders in this analysis that bias estimates of study associations. For example, we controlled for neighborhood-level income, but were unable to include variables like household income, potentially resulting in residual confounding. Second, our study used cross-sectional data to examine mediation, which is an inherently causal and temporal concept that requires the exposure to precede the mediator and the outcome and the mediator to precede the outcome. Regardless, this is one of the first studies to consider mediators in the neighborhood social capital–tooth decay relationship and provides insight for a follow-up longitudinal study that could better ensure proper temporality in the relationships of interest. Third, our study is limited by a small sample size which could be one explanation for statistically insignificant mediation findings. The necessary tools for causal mediation power analyses were not yet developed at the time of data collection for this study, which is a limitation that could be addressed in future work. Fourth, causal mediation analysis requires a number of strong assumptions, including sequential ignorability, which cannot be verified with data. Unfortunately, the sensitivity analyses suggested by Imai et al. to examine the assumption of strong sequential ignorability [45,47] are not available when using the mixed effect models that were needed in this study to account for non-independence at the neighborhood level. Lastly, response rates from neighborhoods were low which means that generalizability of the neighborhood social capital measure may be poor or biased towards individuals from higher response neighborhoods.

This study also has strengths. We adopted theory-driven definitions of social capital, using measures of multiple dimensions of neighborhood social capital. In addition, the analysis was completed under a potential outcomes causal inference framework which advances the quality and application of oral health research findings [60].

## Public health and policy implications

To fully appreciate the broader relevance and findings of this work, it is crucial to consider the shared risk factors of oral health and systemic health, including socioeconomic settings that include but are not limited to the experience of financial, housing, and food insecurity and access to education, employment opportunities, social services, and health care. Further, oral health researchers and practitioners must approach prevention and treatment in an integrative manner acknowledging the contextual determinants of health and the interwoven relationship of oral and systemic health. In terms of public health strategies, social capital can be leveraged by modifying social norms for optimal oral health behaviors or via training for organizing and collective action to improve neighborhood conditions [61].

## Conclusion

In this exploratory study of Medicaid-enrolled adolescents, we found that higher neighborhood social support and informal social control were associated with lower prevalence of worry about food money, a component of food insecurity and known risk factor for poor oral health. However, we did not find that the neighborhood social capital–tooth decay relationship was mediated by worry about food money, which may have resulted from the underpowered sample size. Additional longitudinal studies that can more adequately evaluate potential mediators including food insecurity, oral health routines, and stress, are needed and may eventually be used to develop neighborhood-based interventions and to inform policies that promote the oral health of low-income populations.

## Supporting information

**S1 File.** S1 Table. Survey items used to construct each neighborhood-level social capital measure. S2 Table. Mediating effects of worrying about food money between neighborhood social capital and tooth decay with interaction between neighborhood social capital and worrying about food money for Oregon Medicaid-enrolled adolescents ages 12–18 years who participated in a neighborhood oral health study in 2015–2016 (n = 299). S3 Table. Mediating effects of worrying about food money between neighborhood social capital and DMFS for Oregon Medicaid-enrolled adolescents ages 12–18 years who participated in a neighborhood oral health study in 2015–2016 (n = 299).
(DOCX)

**S1 Data. Social capital oral health dataset n=331.**
(CSV)

**S2 Data. Social capital oral health codebook n=331.**
(XLSX)

## Acknowledgments

We thank the Oregon Health Authority for access to Medicaid data, participating families for their time, and all the private practice dentists in Oregon who welcomed us into their offices to enable clinical data collection.

## Author contributions

**Conceptualization:** Richard M. Carpiano, Adam C. Carle, Kyle Crowder, Donald L. Chi.

**Data curation:** Marilynn Rothen, Michael Yoo, Donald L. Chi.

**Formal analysis:** Courtney M. Hill, Lloyd A. Mancl.

**Funding acquisition:** Donald L. Chi.

**Methodology:** Richard M. Carpiano, Adam C. Carle, Kyle Crowder, Donald L. Chi.

**Writing – original draft:** Courtney M. Hill, Lloyd A. Mancl, Richard M. Carpiano, Adam C. Carle, Marilynn Rothen, Kyle Crowder, Michael Yoo, Donald L. Chi.

**Writing – review & editing:** Courtney M. Hill, Lloyd A. Mancl, Richard M. Carpiano, Adam C. Carle, Marilynn Rothen, Kyle Crowder, Michael Yoo, Donald L. Chi.

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
