## [Decision Letter · Decision Letter 0]

19 Feb 2025

PONE-D-24-43138Associations between neighborhood social capital, oral health risk factors, and tooth decay among Medicaid-enrolled adolescents: a hypothesis-generating preliminary studyPLOS ONE

Dear Dr. Chi,

Thank you for submitting your manuscript to PLOS ONE. After careful consideration, we feel that it has merit but does not fully meet PLOS ONE’s publication criteria as it currently stands. Therefore, we invite you to submit a revised version of the manuscript that addresses the points raised during the review process.

The manuscript is relevant and brings important discussions to public health. Two reviewers have delivered their assessment, and have suggested some points for improving the manuscript. One main point that should be taken into consideration is the understanding of oral health in the context of comprehensive health and how social capital can be increased or improved.

We look forward to receiving your revised manuscript.

Kind regards,

Marília Jesus Batista de Brito Mota, Post-doc

Academic Editor

PLOS ONE

2. Please provide captions for Figure 1 in your manuscript.

Additional Editor Comments:

The manuscript is relevant and brings important discussions to public health. Two reviewers have delivered their assessment, and have suggested some points for improving the manuscript. One main point that should be taken into consideration is the understanding of oral health in the context of comprehensive health and how social capital can be increased or improved.

Reviewers' comments:

Reviewer's Responses to Questions

**Comments to the Author**

1. Is the manuscript technically sound, and do the data support the conclusions?

Reviewer #1: Yes

Reviewer #2: Partly

2. Has the statistical analysis been performed appropriately and rigorously? 

Reviewer #1: Yes

Reviewer #2: Yes

3. Have the authors made all data underlying the findings in their manuscript fully available?

Reviewer #1: Yes

Reviewer #2: No

4. Is the manuscript presented in an intelligible fashion and written in standard English?

Reviewer #1: Yes

Reviewer #2: Yes

5. Review Comments to the Author

Reviewer #1: Review of the manuscript titled "Associations between neighborhood social capital, oral health risk factors, and tooth decay among Medicaid-enrolled adolescents: a hypothesis-generating preliminary study"

The article is well-written and addresses an important topic in public health. It provides a solid theoretical framework and a well-organized structure. A few minor suggestions are offered with the intention of enhancing the text, and I recommend a brief revision to further strengthen the manuscript.

Suggestions:

Introduction

The introduction is well-structured, providing a clear contextual review and a well-defined objective. However, while the study's aim is clear, it could be emphasized more directly. I suggest reformatting the final paragraph to more explicitly highlight the study’s contribution.

Methodology and Results

The methodology is well-described, presented clearly and in detail, and supported by references to studies that utilized similar variables. This allows readers to understand how the study was conducted and the evidence base supporting the methodological choices. However, regarding the results description, my suggestion is that the authors focus more on interpreting the findings, rather than merely replicating the results from the tables. For instance, when presenting the finding "support had a lower prevalence of worrying about food money (prevalence ratio [PR] 0.74; 95% CI 0.56, 0.96; p=.02)," it would be helpful to explain what this really means. This will help contextualize the finding, and it would be even more enriching if the authors could explain, in an accessible manner, what this 26% reduction represents in the context of the study population.

Discussion

The discussion is well-structured and addresses important aspects. However, it is crucial for the authors to consider adding a paragraph that discusses a more integrated perspective, taking into account shared risk factors such as food insecurity, stress, and socio-economic conditions, which affect not only oral health but also other aspects of overall health. Oral health should be approached within a broader context that recognizes the interaction between different social determinants and the need for intersectoral public policies. This point could be highlighted in the "Strengths and Limitations" section or following it, emphasizing that, while social capital plays an important role, it needs to be understood within a broader set of health determinants such as education, income, and access to services.

Additionally, how could social capital be integrated into public health strategies? This should also be addressed more thoroughly in the discussion.

Reviewer #2: Title: Associations between neighborhood social capital, oral health risk factors, and tooth decay among Medicaid-enrolled adolescents: a hypothesis-generating preliminary study

A relatively recent research topic, the article is relevant when bringing together neighborhood, adolescents and oral health - using mixed model effects. However, it does not bring significant news from a scientific point of view and presents relevant methodological limitations.

Abstract

Considering Medicad as a specific US program, I suggest explaining it in the abstract.

Introduction

Page 4 Line 29 – “Atleast half of the U.S. population aged 12-19 years has treated or untreated tooth decay” It is necessary to explain when this data was presented.

Page 5 Line 61 - If there is a systematic review on social capital and oral health in adolescents, the need for another study is not clear.

I also suggest adding the study by Silva et al. 2025 - da Silva PM, Knorst JK, Ardenghi TM, Tomazoni F. Relationship among different levels of individual and community social capital with oral health related-quality of life among adolescents. J Dent. 2025 Jan;152:105487. doi: 10.1016/j.jdent.2024.105487. Epub 2024 Dec 3. PMID: 39626348.

Methods

The Methods needs to be rewritten. There is a lot of repeated information. Example, Line 130 and Line 137; Line 137 and Line 165.

I suggest following the STROBE (https://www.strobe-statement.org/) to describe. Information is repeated and inconsistent.

Page 7 Line 112 – “Our multilevel study”. Is it a multilevel or multicentric study?

Data collection was carried out in 2015 to 2016. Wouldn't the results achieved be outdated today (10 years later)?

The data analysis method is robust, a fact that adds value to the research in question.

Results

There are variables shown in results that are not shown in methods, for example (line 279) " 24% had irregular dental care".

How was family routine assessed and measured? It's not clear.

I suggest presenting the result of the GLMM model in figure.

Considering the magnitude of the research and the high number of variables, it is necessary to review methods and results. Describe in more detail the materials and methods. Focus on the objective.

Discussion

Page 22 Line 340 - “The connection between social support and food insecurity has been previously reported in the literature29-31,51”. Considering this the main result of the study, are there advances in the scientific literature with the results presented?

Avoid repeating results information in the discussion. For example, Line 351-352.

Conclusion

Page 28 Line 460 - “Oral health disparities persist in the U.S. and social determinants like social capital may be an important risk factor to target in future oral health interventions”. Is this conclusion supported by your results?

The conclusion should only respond to the objective.

6. PLOS authors have the option to publish the peer review history of their article (what does this mean? ). If published, this will include your full peer review and any attached files.

**Do you want your identity to be public for this peer review?** For information about this choice, including consent withdrawal, please see our Privacy Policy .

Reviewer #1: **Yes: ** Professor. Orlando Luiz do Amaral Júnior

Reviewer #2: No

---

## [Author Response · Author response to Decision Letter 1]

15 Apr 2025

RESPONSE: All PLOS ONE style requirements are met.

2. Please provide captions for Figure 1 in your manuscript.

RESPONSE: The figure caption was added to the manuscript after the text but before the references.

Additional Editor Comments:

The manuscript is relevant and brings important discussions to public health. Two reviewers have delivered their assessment, and have suggested some points for improving the manuscript. One main point that should be taken into consideration is the understanding of oral health in the context of comprehensive health and how social capital can be increased or improved.

Reviewers' comments:

Reviewer's Responses to Questions

Comments to the Author

1. Is the manuscript technically sound, and do the data support the conclusions?

Reviewer #1: Yes

Reviewer #2: Partly

2. Has the statistical analysis been performed appropriately and rigorously?

Reviewer #1: Yes

Reviewer #2: Yes

3. Have the authors made all data underlying the findings in their manuscript fully available?

RESPONSE: We can do this – need to restrict dataset to the variables that we analyzed and make sure it’s completely de-identified.

Reviewer #1: Yes

Reviewer #2: No

4. Is the manuscript presented in an intelligible fashion and written in standard English?

Reviewer #1: Yes

Reviewer #2: Yes

5. Review Comments to the Author

RESPONSE: We appreciate the reviewers’ thoughtful comments and feedback on our manuscript. Below we provide responses to queries and suggestions posed by the reviewers. We appreciate the opportunity to improve upon our manuscript.

Reviewer #1: Review of the manuscript titled "Associations between neighborhood social capital, oral health risk factors, and tooth decay among Medicaid-enrolled adolescents: a hypothesis-generating preliminary study"

The article is well-written and addresses an important topic in public health. It provides a solid theoretical framework and a well-organized structure. A few minor suggestions are offered with the intention of enhancing the text, and I recommend a brief revision to further strengthen the manuscript.

Suggestions:

Introduction

The introduction is well-structured, providing a clear contextual review and a well-defined objective. However, while the study's aim is clear, it could be emphasized more directly. I suggest reformatting the final paragraph to more explicitly highlight the study’s contribution.

RESPONSE: We highlighted the contribution of the study in the final paragraph of the introduction: “The study contributes to efforts examining social capital as a determinant of oral health and aims to identify potential mediators of the relationship.” (lines 98-100)

Methodology and Results

The methodology is well-described, presented clearly and in detail, and supported by references to studies that utilized similar variables. This allows readers to understand how the study was conducted and the evidence base supporting the methodological choices. However, regarding the results description, my suggestion is that the authors focus more on interpreting the findings, rather than merely replicating the results from the tables. For instance, when presenting the finding "support had a lower prevalence of worrying about food money (prevalence ratio [PR] 0.74; 95% CI 0.56, 0.96; p=.02)," it would be helpful to explain what this really means. This will help contextualize the finding, and it would be even more enriching if the authors could explain, in an accessible manner, what this 26% reduction represents in the context of the study population.

RESPONSE: We added additional text to the results to expand on what a prevalence ratio of 0.74 means: “This means that households in neighborhoods with higher social support reported on average were 26% less likely to experience worry about food money (a form of food insecurity).” (lines 296-298)

Discussion

The discussion is well-structured and addresses important aspects. However, it is crucial for the authors to consider adding a paragraph that discusses a more integrated perspective, taking into account shared risk factors such as food insecurity, stress, and socio-economic conditions, which affect not only oral health but also other aspects of overall health. Oral health should be approached within a broader context that recognizes the interaction between different social determinants and the need for intersectoral public policies. This point could be highlighted in the "Strengths and Limitations" section or following it, emphasizing that, while social capital plays an important role, it needs to be understood within a broader set of health determinants such as education, income, and access to services. Additionally, how could social capital be integrated into public health strategies? This should also be addressed more thoroughly in the discussion.

RESPONSE: We added a paragraph after strength/limitations that discusses an integrated perspective to promote oral health and potential strategies to address social capital:

“Public Health and Policy Implications

To fully appreciate the relevance and findings of this work, it is crucial to consider the shared risk factors of oral health and systemic health, including socioeconomic settings that include but are not limited to the experience of financial, housing, and food insecurity and access to education, employment opportunities, social services, and health care. Further, oral health researchers and practitioners must approach prevention and treatment in an integrative manner acknowledging the contextual determinants of health and the interwoven relationship of oral and systemic health. In terms of public health strategies, social capital can be leveraged by modifying social norms for optimal oral health behaviors or via training for organizing and collective action to improve neighborhood conditions.61” (lines 463-472).

The introduction also describes the role social capital in improving community advocacy: “Similarly, in a neighborhood with high levels of organization participation, community advocacy efforts can secure better resources in the neighborhood like access to healthier food stores, fluoridated tap water, and community clinics that provide comprehensive dental care services. In neighborhoods with high informal social control, adherence to social norms that include healthy behaviors (i.e., healthy eating versus sugary snacks and fast food family; routines like eating meals together every day; or enforcement of oral hygiene behaviors like regular tooth brushing and dental care) may also improve oral health outcomes.” (lines 82-89)

Reviewer #2: Title: Associations between neighborhood social capital, oral health risk factors, and tooth decay among Medicaid-enrolled adolescents: a hypothesis-generating preliminary study

A relatively recent research topic, the article is relevant when bringing together neighborhood, adolescents and oral health - using mixed model effects. However, it does not bring significant news from a scientific point of view and presents relevant methodological limitations.

Abstract

Considering Medicad as a specific US program, I suggest explaining it in the abstract.

RESPONSE: We added text to the abstract to state that “Medicaid is a public insurance program in the U.S. providing dental insurance to low-income children” (line 9-10).

Introduction

Page 4 Line 29 – “Atleast half of the U.S. population aged 12-19 years has treated or untreated tooth decay” It is necessary to explain when this data was presented.

RESPONSE: The data presented are from 2018 analyses of nationally representative 2015-2016 data: “At least one-half of the U.S. population aged 12-19 years has treated or untreated tooth decay, according to the most recent national estimates from 2015-2016.5” (line 31).

Page 5 Line 61 - If there is a systematic review on social capital and oral health in adolescents, the need for another study is not clear.

I also suggest adding the study by Silva et al. 2025 - da Silva PM, Knorst JK, Ardenghi TM, Tomazoni F. Relationship among different levels of individual and community social capital with oral health related-quality of life among adolescents. J Dent. 2025 Jan;152:105487. doi: 10.1016/j.jdent.2024.105487. Epub 2024 Dec 3. PMID: 39626348.

RESPONSE: The cited systematic review includes 7 studies that examined social capital at a contextual level, of which 4 included objective assessments of oral health. Even after critically reviewing the results of the systematic review, there are a number of questions which warrant further investigation. For example, what is the appropriate level at which to assess social capital (individual vs. contextual)? In addition, how should social capital be measured? Most studies in the review paper used only indicators or proxies that were hypothesized to be related to social capital rather than multiple dimensions of neighborhood social capital developed via theory like in our analysis. Our study also aims to further identify potential neighborhood, household, and individual factors that could mechanistically explain an association between social capital and oral health.

We incorporated the Silva 2025 study in the introduction: “…and lack of individual or community social capital was associated with lower oral health related quality of life.16” (lines 61-63)

Methods

The Methods needs to be rewritten. There is a lot of repeated information. Example, Line 130 and Line 137; Line 137 and Line 165.

I suggest following the STROBE (https://www.strobe-statement.org/) to describe. Information is repeated and inconsistent.

RESPONSE: We removed the sentences from the “adolescent data collection” section that referenced the period of data collection and the specific measures that were collected from adolescents to avoid repeated information.

Page 7 Line 112 – “Our multilevel study”. Is it a multilevel or multicentric study?

RESPONSE: A multilevel study typically refers to a study in which there are variables assessed at multiple levels and oftentimes when group levels (or units) are at a lower level nested within units at a higher level (Diez Roux 2000). Our study is multilevel because we collected data at multiple levels – individual (oral health), household (routines, food insecurity), and neighborhood (social capital). We added the Diez Roux reference to our manuscript for clarity (line 115).

Data collection was carried out in 2015 to 2016. Wouldn't the results achieved be outdated today (10 years later)?

RESPONSE: It is possible that certain aspects of the neighborhoods and individuals under study have changed. However, the overall goal of the manuscript was to study how social capital relates to oral health among adolescents and the underlying social, economic, and biologic mechanisms that explain that relationship. These should not have fundamentally changed since the time of data collection.

The data analysis method is robust, a fact that adds value to the research in question.

Results

There are variables shown in results that are not shown in methods, for example (line 279) " 24% had irregular dental care".

RESPONSE: Notes under Table 1 state that, “To measure whether adolescents had irregular dental care use, we asked caregivers on the caregiver survey, “Does your child currently see a dentist for regular checkups?” A “no” response was considered irregular dental care.” For clarity, we added this to the revised manuscript text (lines 233-236).

How was family routine assessed and measured? It's not clear.

RESPONSE: The manuscript text states that family routines were assessed using the family routines index (FRI): “Two types of routines were assessed: inconsistent general family routines and inconsistent oral health routines. We measured the former using a modified version of the validated Family Routine Inventory (FRI) which measures cohesion, solidarity, order, and overall satisfaction with family life.39 The FRI contains 26 items. Three items were not applicable to adolescents (e.g., “parents read or tell stories to the children almost every day”) and thus removed. Response options were almost never, 1-2 times a week, 3-5 times a week, always/everyday. A composite score was created by summing the responses for the remaining 23 items and was reverse-coded so larger values indicated lower endorsement and adherence to a routine. The range of possible scores was 0 to 69 and missing item-level data were counted as 0.” (lines 188-196).

I suggest presenting the result of the GLMM model in figure.

RESPONSE: Typically, regression results are reported in a table. If primary predictor variables were categorical, we could use a figure to display the estimated marginal means. However, in our manuscript, the social capital measures are continuous, so this is not an option.

Considering the magnitude of the research and the high number of variables, it is necessary to review methods and results. Describe in more detail the materials and methods. Focus on the objective.

RESPONSE: We reviewed the methods and results sections carefully editing for clarity, removing repeat information, and ensuring that descriptions were consistent. A few small edits were made to remove extraneous text, describe additional details, and better organize text.

Discussion

Page 22 Line 340 - “The connection between social support and food insecurity has been previously reported in the literature29-31,51”. Considering this the main result of the study, are there advances in the scientific literature with the results presented?

Avoid repeating results information in the discussion. For example, Line 351-352.

RESPONSE: Our study not only demonstrated the connection between social support and food insecurity, but provided evidence to the hypothesis that experience of household food insecurity could be one mechanism by which a social capital exerts influence on oral health. We removed the line that summarized the finding about food insecurity and tooth decay: “Living in a household that worried about food money often was associated with 2.6 times the decayed tooth su

---

## [Decision Letter · Decision Letter 1]

30 Jun 2025

PONE-D-24-43138R1Associations between neighborhood social capital, oral health risk factors, and tooth decay among Medicaid-enrolled adolescents: a hypothesis-generating preliminary studyPLOS ONE

Dear Dr. Chi,

Thank you for submitting your manuscript to PLOS ONE. After careful consideration, we feel that it has merit but does not fully meet PLOS ONE’s publication criteria as it currently stands. Therefore, we invite you to submit a revised version of the manuscript that addresses the points raised during the review process.

The manuscript "Associations between neighborhood social capital, oral health risk factors, and tooth decay among Medicaid-enrolled adolescents: a hypothesis-generating preliminary study" has been reviewed and the authors have followed the reviewers' recommendations, although some points still need to be clarified in order to improve the manuscript. To this end, I recommend a minor revision of the manuscript, adding the reviewer's suggestions.

We look forward to receiving your revised manuscript.

Kind regards,

Marília Jesus Batista de Brito Mota, Post-doc

Academic Editor

PLOS ONE

Journal Requirements:

Additional Editor Comments (if provided):

The manuscript "Associations between neighborhood social capital, oral health risk factors, and tooth decay among Medicaid-enrolled adolescents: a hypothesis-generating preliminary study" has been reviewed and the authors have followed the reviewers' recommendations, although some points still need to be clarified in order to improve the manuscript. To this end, I recommend a new revision of the manuscript, adding the reviewer's suggestions.

Reviewers' comments:

Reviewer's Responses to Questions

**Comments to the Author**

1. If the authors have adequately addressed your comments raised in a previous round of review and you feel that this manuscript is now acceptable for publication, you may indicate that here to bypass the “Comments to the Author” section, enter your conflict of interest statement in the “Confidential to Editor” section, and submit your "Accept" recommendation.

Reviewer #2: All comments have been addressed

Reviewer #3: All comments have been addressed

2. Is the manuscript technically sound, and do the data support the conclusions?

Reviewer #2: Yes

Reviewer #3: Yes

3. Has the statistical analysis been performed appropriately and rigorously? 

Reviewer #2: Yes

Reviewer #3: Yes

4. Have the authors made all data underlying the findings in their manuscript fully available?

Reviewer #2: Yes

Reviewer #3: Yes

5. Is the manuscript presented in an intelligible fashion and written in standard English?

Reviewer #2: Yes

Reviewer #3: Yes

6. Review Comments to the Author

Reviewer #2: I consider all the answers presented to be pertinent. The changes made to the manuscript are appropriate.

Reviewer #3: Manuscript Title: Associations between neighbourhood social capital, oral health risk factors, and tooth decay among Medicaid-enrolled adolescents: a hypothesis-generating preliminary study

Summary

This manuscript explores the association between neighbourhood social capital and untreated tooth decay among Medicaid-enrolled adolescents, with mediation analyses examining how food insecurity and other oral health risk factors might explain this relationship. The revised version addresses key reviewer comments from the initial submission, with expanded explanations, clarified methodology, and improved discussion.

1. Technical Soundness and Statistical Rigor

The study uses appropriate statistical methods, including GLMM and causal mediation analysis based on the potential outcomes’ framework. The authors correctly account for clustering and adjust for key confounders. However, a few limitations remain:

i. The sample size (n = 299) limits statistical power, particularly for detecting mediation effects.

ii. The cross-sectional design restricts causal inference, which the authors now appropriately acknowledge.

iii. The neighbourhood-level social capital measures, though theoretically grounded, may be subject to nonresponse bias due to low informant participation (mean response rate: 11%).

Assessment: Technically sound, but interpretation should be appropriately cautious due to data and design constraints.

2. Revisions and responsiveness to reviewer feedback

The authors have been thorough in responding to comments:

i. Clarified the role and relevance of Medicaid in the abstract.

ii. Enhanced the description and justification of the conceptual model and statistical methods.

iii. Added context for key findings (e.g., interpreting a 26% reduction in food insecurity).

iv. Introduced a more holistic view of oral health as part of general health within a social determinant’s framework.

v. Added the suggested citation (Silva et al. 2025) and addressed redundancy in the Methods section.

These revisions improve the clarity, relevance, and policy orientation of the manuscript.

3. Remaining issues and suggestions

i. Generality of the results: While the study focuses on Medicaid-enrolled adolescents in Oregon, results may not be generalizable to other populations. The authors could briefly emphasize this in the discussion.

ii. Low power of mediation analysis: Though acknowledged, it would help if the authors explicitly stated that observed null indirect effects should be interpreted cautiously due to underpowering.

iii. Figure for GLMM results: Reviewer 2 suggested visualizing GLMM results. Although authors explain that figures may not be ideal for continuous predictors, a graphical summary (e.g., caterpillar plot of PRs with CIs) could still aid reader interpretation.

The authors have improved the manuscript significantly and are transparent about limitations.

7. PLOS authors have the option to publish the peer review history of their article (what does this mean? ). If published, this will include your full peer review and any attached files.

**Do you want your identity to be public for this peer review?** For information about this choice, including consent withdrawal, please see our Privacy Policy .

Reviewer #2: No

Reviewer #3: **Yes: ** Martina Mchenga

---

## [Author Response · Author response to Decision Letter 2]

1 Jul 2025

3. Remaining issues and suggestions

i. Generality of the results: While the study focuses on Medicaid-enrolled adolescents in Oregon, results may not be generalizable to other populations. The authors could briefly emphasize this in the discussion.

RESPONSE: We revised the discussion to acknowledge and comment on the generalizability of the findings: “However, further research is needed on populations beyond Medicaid-enrolled adolescents to evaluate the generalizability of these findings.” (lines 358-360).

ii. Low power of mediation analysis: Though acknowledged, it would help if the authors explicitly stated that observed null indirect effects should be interpreted cautiously due to underpowering.

RESPONSE: We added a statement to the discussion to state that the null indirect effects should be interpreted with caution due to limited power“…and the null findings should be interpreted cautiously due to our underpowered sample size” (line 365-366).

iii. Figure for GLMM results: Reviewer 2 suggested visualizing GLMM results. Although authors explain that figures may not be ideal for continuous predictors, a graphical summary (e.g., caterpillar plot of PRs with CIs) could still aid reader interpretation.

RESPONSE: In table 2, our statistical modeling included log-binomial regression for prevalence ratios (PRs) and linear regression for beta coefficients, resulting in estimates on different scales (multiplicative vs. additive), which complicates graphical summarization. Additionally, Table 3 presents only six PRs, which we believe are sufficiently interpretable without a figure. Thus, we opted not to include a graphical summary, as it may not add clarity in this context.

The authors have improved the manuscript significantly and are transparent about limitations.

RESPONSE: Thank you for reviewing our manuscript and providing comments to strengthen it.

---

## [Decision Letter · Decision Letter 2]

23 Jul 2025

Associations between neighborhood social capital, oral health risk factors, and tooth decay among Medicaid-enrolled adolescents: a hypothesis-generating preliminary study

PONE-D-24-43138R2

Dear Dr. Chi,

We’re pleased to inform you that your manuscript has been judged scientifically suitable for publication and will be formally accepted for publication once it meets all outstanding technical requirements.

Kind regards,

Marília Jesus Batista de Brito Mota, Post-doc

Academic Editor

PLOS ONE

Additional Editor Comments (optional):

All comments have been address. Just the two minor suggestions before final publication.

1. Response-rate limitation – Low neighbourhood survey response (mean 11 %) is acknowledged in Limitations but could be flagged earlier in Methods sampling subsection for transparency.

2.Typographic clean-up - A final proof-read to catch occasional “saple” (p.9) and stray double spaces.

Reviewers' comments:

Reviewer's Responses to Questions

**Comments to the Author**

1. If the authors have adequately addressed your comments raised in a previous round of review and you feel that this manuscript is now acceptable for publication, you may indicate that here to bypass the “Comments to the Author” section, enter your conflict of interest statement in the “Confidential to Editor” section, and submit your "Accept" recommendation.

Reviewer #3: All comments have been addressed

2. Is the manuscript technically sound, and do the data support the conclusions?

Reviewer #3: Yes

3. Has the statistical analysis been performed appropriately and rigorously? 

Reviewer #3: Yes

4. Have the authors made all data underlying the findings in their manuscript fully available?

Reviewer #3: Yes

5. Is the manuscript presented in an intelligible fashion and written in standard English?

Reviewer #3: Yes

6. Review Comments to the Author

Reviewer #3: All comments have been address. Just the two minor suggestions before final publication.

1. Response-rate limitation – Low neighbourhood survey response (mean 11 %) is acknowledged in Limitations but could be flagged earlier in Methods sampling subsection for transparency.

2.Typographic clean-up - A final proof-read to catch occasional “saple” (p.9) and stray double spaces.

7. PLOS authors have the option to publish the peer review history of their article (what does this mean? ). If published, this will include your full peer review and any attached files.

**Do you want your identity to be public for this peer review?** For information about this choice, including consent withdrawal, please see our Privacy Policy .

Reviewer #3: **Yes: ** Martina Mchenga
